# Zinc in the Brain: Friend or Foe?

**DOI:** 10.3390/ijms21238941

**Published:** 2020-11-25

**Authors:** Seunghyuk Choi, Dae Ki Hong, Bo Young Choi, Sang Won Suh

**Affiliations:** Department of Physiology, College of Medicine, Hallym University, Chuncheon 24252, Korea; bmchoi311@hallym.ac.kr (S.C.); zxnm01220@gmail.com (D.K.H.); bychoi@hallym.ac.kr (B.Y.C.)

**Keywords:** zinc, physiology, pathophysiology, brain

## Abstract

Zinc is a trace metal ion in the central nervous system that plays important biological roles, such as in catalysis, structure, and regulation. It contributes to antioxidant function and the proper functioning of the immune system. In view of these characteristics of zinc, it plays an important role in neurophysiology, which leads to cell growth and cell proliferation. However, after brain disease, excessively released and accumulated zinc ions cause neurotoxic damage to postsynaptic neurons. On the other hand, zinc deficiency induces degeneration and cognitive decline disorders, such as increased neuronal death and decreased learning and memory. Given the importance of balance in this context, zinc is a biological component that plays an important physiological role in the central nervous system, but a pathophysiological role in major neurological disorders. In this review, we focus on the multiple roles of zinc in the brain.

## 1. Introduction

The trace metal ion zinc is one of the most prevalent and essential elements that are involved in brain function, and it plays a role in both physiological and pathophysiological processes. Neurons containing “free ionic zinc” (Zn^2+^) are found in various areas of the brain, including the cortex, amygdala, olfactory bulb, and hippocampal neurons, which appear to have the highest concentration of zinc in the brain. Zinc has been implicated in the biological activity of enzymes, proteins, and signal transcription factors, as well as in the maintenance of various homeostatic mechanisms, acting as structural, regulatory, and catalytic cofactors for a variety of enzymes, such as DNA and RNA polymerases, histone deacetylases, and DNA ligases. Zinc is also important for cell growth and genomic stability [1,2,3,4,5].

In its role as a neuromodulator, zinc is released during synaptic transmission and it binds to presynaptic or postsynaptic membrane receptors; as such, it can translocate from presynaptic terminals to postsynaptic neurons [6,7]. Zinc is colocalized in the synaptic vesicles of glutamatergic neurons. Therefore, zinc is released from glutamatergic synaptic vesicles and it then reacts with excitatory amino acid receptors N-methyl-D aspartic acid (NMDA) and α-amino-3-hydroxy-5-methyl-4-isoxazolepropionic acid (AMPA) and the inhibitory amino acid receptors and γ-aminobutyric acid (GABA) [8,9,10]. Besides, extracellular Zn^2+^ can alter neuronal excitability due to its effects on various voltage-gated ion channels [11,12,13].

As a neuromodulator, zinc performs various physiological functions. Zinc is essential for regulating cell proliferation in several ways, including the hormonal regulation of cell division. In addition, zinc represents a molecular signal to immune cells and transcription factors that are involved in the expression of inflammatory cytokines. Zinc supplementation studies have shown decreased infection rates and inflammatory cytokine production. Furthermore, zinc has a metal binding ability, and it is well known for its antioxidant properties. Zinc is a redox inert metal that acts as an antioxidant through the catalysis of copper/zinc-superoxide dismutase, the protection of the protein sulfhydryl group, and the upregulation of metallothionein (MT) expression [14,15].

However, the other, less physiological, characteristic of zinc can cause cell death at high concentrations in neurons. An accumulation of zinc can lead to neuronal death following a variety of brain insults (stroke, epileptic seizures, hypoglycemia, and trauma injury). In addition, zinc deficiency induces apoptotic neuronal cell death through the intrinsic mitochondrial pathway, which can be triggered by the activation of caspase-3 and the abnormal regulation of pro-survival pathways (extracellular signal-regulated kinase (ERK1/2), nuclear factor-κB (NF-κB)) [11,16,17]. In this review, the recent progress in our understanding of the beneficial physiological—as well as deleterious pathophysiological—roles of zinc in the brain is summarized.

## 2. Friend

The important role that zinc plays in the body was recognized by observing its effect on several biochemical and physiological functions. In particular, zinc plays a structural or functional role for numerous proteins. Zinc regulates the activity of proteins, such as receptors and enzymes that are involved in the regulation of numerous processes, including macromolecular synthesis, the regulation of signal transduction and gene transcription, and transport processes. Zinc has recently been recognized as a second messenger that is involved in intracellular signaling. Zinc is also involved in maintaining genomic stability through several actions, including redox homeostasis regulation, DNA repair, synthesis, and methylation. Zinc also serves as a molecular signal for immune cells [14,18,19]. Zinc regulates various transcription factors that regulate gene expression and that are involved in inflammatory cytokine and adhesion molecule signaling pathways.

### 2.1. Zinc Is Essential for Cell Growth

Zinc was first recognized as essential for growth (Figure 1) in rats [20]. Zinc participates in various cellular processes, acting as a cofactor for many enzymes, and it affects gene expression through transcription factor regulation. The relationship between zinc in DNA and protein synthesis is direct and evident. Zinc is present in the nucleus, nucleolus, and chromosomes. Zinc stabilizes the structure of RNA, DNA, and ribosomes [21]. Zinc metalloenzymes are associated with various DNA or RNA synthesis enzymes, such as RNA polymerase, reverse transcriptase, and transcription factor IIIA [21,22].

A common structural motif showing that the zinc ion plays a key role in protein complexes is the zinc finger domain. The zinc finger domain is a structure in which zinc ions form a bridge between cysteine and histidine residues in order to form a polypeptide chain. Many proteins have been discovered that contain zinc fingers, and this motif is considered to be one of the three that are essential for eukaryotic regulatory proteins to bind specific DNA sequences [23].

The pituitary gland has the highest concentration of zinc of any organ, and zinc also enhances the function of pituitary hormones [24]. Because the pituitary gland is a source of growth hormone (GH), it is a major endocrine regulator of somatic cell growth. Root et al. (1979) and several reports have investigated the role of GH in the suppression of growth by zinc deficiency. Zinc deficiency causes pituitary GH secretion deficiency [25]; additionally, zinc-deficient rats showed reduced circulating GH concentrations [26].

Zinc also affects the regulation of cell division growth factors, such as insulin-like growth factor-1 (IGF-1) or nerve growth factor (NGF). IGF-I mediates a variety of cellular processes, including the stimulation of amino acid, glucose uptake, and the regulation of the cell cycle. It binds to membrane-associated receptors with tyrosine kinase activity [27]. Cossack et al. (1991) reported that decreased serum IGF-1 occurs in humans and animals when dietary energy or protein intake is inadequate. In humans, zinc deficiency reduces circulating IGF-1 levels independent of the total energy intake levels [28]. A decrease in IGF-1 corresponds to a decrease in serum zinc concentration [29]. Cell division by growth factors is regulated by a mechanism in which the ligand activates intracellular signaling pathways by binding to its cognate receptor, which activates intracellular signaling pathways. The IGF-1 receptor possesses a unique tyrosine kinase that is thought to initiate the phosphorylation cascade [15,30,31].

Zinc is a component of the nerve growth factor-7S (NGF-7S) molecule. The hippocampal region is rich in neurotrophic factors, and the cholinergic deafferentation of the hippocampus appears to activate these neurotrophic factors [32]. NGF is a founding member of the neurotrophin family, which is a family of secreted growth factors that are involved in the growth, survival, and developmental plasticity of neuronal populations in vertebrate peripheral and central nervous systems [33]. NGF’s selective receptor TrkA is highly expressed by nociceptors, showing that NGF plays an important role in nociception. In the absence of NGF, neurons undergo protein synthesis-dependent apoptosis [34,35,36].

The Zn^2+^-sensing G-protein coupled receptor (ZnR/GPR39) has been shown to regulate the activity of ion transport mechanisms that are important for the physiological function of epithelial and neuronal cells [37]. In particular, ZnR/GPR39 activity enhanced neuronal inhibitory tone. Zinc deficiency has been linked to epilepsy and seizures, which suggests an important physiological role for ZnR/GPR39 [38,39,40]. This specific extracellular receptor has been strongly linked to the regulation of the mitogen-activated protein kinase (MAPK) and PI3K pathways [41,42,43,44]. Additionally, it triggers the activation of ERK1/2, which, in turn, activates p21 and cyclin D1 [45]. The activation of ZnR/GPR39 by Zn^2+^ also upregulates K^+^/Cl^−^ co-transport (KCC) activity. These ZnR/GPR39 and KCC inputs are essential for mediating Zn^2+^-dependent cellular signaling pathways that lead to cell survival, migration, and proliferation [46].

Therefore, zinc is an essential component of cell growth that mediates the activity of growth hormones and promotes DNA synthesis.

### 2.2. Zinc Increases Neurogenesis

Many previous studies have identified numerous important roles of zinc on the process of neurogenesis [2]. Several investigators have found mechanisms of neural proliferation under zinc supplementation or zinc-deficient conditions.

The postnatal rat cerebellum with zinc deficiency shows decreased neurogenesis and a decreased expression of genes that are related to hippocampal proliferation and neuronal differentiation [47,48]. Furthermore, in cultured cell experiments, zinc deficiency impairs the proliferation of human neuroblastoma cell line (IMR-32), induces apoptosis, and inhibits neuronal differentiation by retinoic acid [1,47].

Zinc is involved in the regulation of cell growth via several pathways. It is essential for the enzymatic system that governs cell division and proliferation. Zinc appears to be essential for the induction of cell proliferation by pituitary growth hormone (GH) and IGF-1. IGF-1 is one of the most critical activators of the protein kinase B (Akt) signaling pathway and it stimulates cell growth and proliferation. The mediation of cell division by growth factors requires the binding of a ligand to its cognate receptor, which activates intracellular signaling pathways [15,30,31] (Figure 1).

Growth factor-activated Akt signaling acts on various downstream factors that are involved in promoting G1/S and G2/M transition; it facilitates progression throughout the normal and unperturbed cell cycle [49]. Activated Akt kinase directly/indirectly regulates various targeted proteins of Cyclins and Cdks that are involved in the function of cell cycle progression of G1/S and G2/M transition [50,51,52,53,54,55]. Zinc deficiency can result in the inhibition of neuronal cell proliferation secondary to an arrest at the G0/G1 phase of the cell cycle, which induces apoptosis in IMR-32 and neuroblastoma cells. In proliferating neurons, zinc deficiency triggers neuronal death via the intrinsic apoptotic pathway [1].

The transcription factors NF-κB, nuclear factor of activated T-cells (NFAT), and activator protein-1 (AP-1) play important roles in brain development by regulating the expression of genes that control processes, such as cell proliferation, differentiation, and synaptic plasticity [56,57,58,59,60]. Aimo et al. (2010) suggested that nuclear factor-κB (NF-κB) and NFAT–DNA binding in nuclear fractions was significantly lower in marginal zinc diet (MZD) brains than in controls [61]. Additionally, the DNA binding factor of NF-κB and nuclear factor of activated T-cells was decreased by both severe and marginal deficiency [61]. NFAT is expressed in neurogenic regions, such as the subventricular zone (SVZ) and hippocampus, where one or more NFAT isoforms may be expressed in neural progenitor cells (NPCs). The inhibition of NFAT activity reduces NPC proliferation [62]. NF-κB is a transcription factor that plays an important role in regulating the expression of genes that are involved in various cellular processes, including innate and adaptive immune responses, cell survival, and proliferation [63,64].

Pregnant rats that were fed a marginally zinc-deficient diet during pregnancy produced E19 embryos with reduced neural progenitor cell proliferation in the ventricular zone. These effects were due to a decrease in extracellular signal-regulated kinase 1/2 (ERK1/2) phosphorylation following a decrease in the zinc inhibition of an ERK1/2-targeting phosphatase, protein phosphatase 2A [65]. The ERK1/2 pathway is also a major regulator of NPC proliferation and neuronal differentiation [66]. Follow-up studies show that neuronal loss persists through adulthood and zinc deficiency regulates key transcription factors for neuronal development, such as Sox2, Pax6, Tbr1, and Tbr2, in addition to changes in ERK1/2 signaling [67]. Similarly, in marginal zinc deficiency (MZD), the fetal brain showed high levels of activation of upstream mitogen-activated protein kinase (MAPK), JNK, and p38. Low levels of ERK phosphorylation were also observed [65,66]. Zinc supplementation promoted a high proliferation rate and differentiation into early neurons or neuron-like cells while using adipose-derived mesenchymal stem cells (AD-MSCs). Furthermore, the chelation of zinc inhibited zinc-supplemented AD-MSC proliferation through the downregulation of ERK1/2 activity [5].

In zinc-related adult neurogenesis, a study of the role of zinc in the regulation of adult neural stem cell proliferation has shown that a zinc-deficient diet reduced the number of cells by half [2]. Additionally, zinc deficiency followed by rescue with adequate levels of dietary zinc in human neural progenitor cells (NT-2) showed p53-mediated alterations in the expression of genes that are involved in the zinc regulation of the G1 and S phases of the cell cycle [3].

Zinc transporter 3 (ZnT3) is the sole route by which zinc ions are loaded into synaptic vesicles of a subset of glutamate neurons in the brain. Vesicular zinc is released into the synapse in an activity-dependent manner, which exerts many signaling functions [68]. ZnT3 knockout mice are deficient in total zinc concentrations at both three and six months. ZnT3 is known to be heavily involved in learning and memory and also to play a major role in aging-related neurodegenerative diseases [69]. Our laboratory has demonstrated that synaptic zinc is an important factor for modulating hippocampal neurogenesis, and we used ZnT3 knockout mice to address this question. ZnT3 deficiency reduced progenitor cell proliferation and neuronal differentiation in the hippocampus, which was mediated by decreasing levels of IGF-1, ERK1/2 phosphorylation, and cyclic AMP response element binding protein (CREB) in the hippocampus of ZnT3 knockout mice at three months of age [70].

ZnT3 knockout mice showed increased basal dendrite length of layer 2/3 pyramidal neurons of the barrel cortex, as compared to wild-type mice [71].

ZnT3 is involved in the regulation of hippocampal neurogenesis, as numerous previous studies have described. In particular, ZnT3 KO mice show abnormalities in adult hippocampal neurogenesis compared with WT mice in response to hypoglycemia [4]. Some research groups have performed interesting studies on how neurogenesis is regulated by environment and stress. In particular, neurogenesis can be enhanced in rodents by transferring environments (from standard laboratory housing to more complex, enriched environments) and also by the application of serotonin reuptake inhibitors (SSRI) [72,73]. However, ZnT3 KO mice do not show improved neurogenesis in response to SSRI treatment [74] or the benefits of enriched housing [75].

McAllister et al. (2020) confirmed that ZnT3 KO mice under RSD stress may regulate neurogenesis through an increased survival rate of newly born cell, similar to WT mice under RSD [76]. However, there is still no evidence of changes in ZnT3 behavior by fluoxetine treatment. Therefore, it partially supports the previous findings [77,78,79] that vesicular zinc is required for the neurogenic effects of chronic fluoxetine treatment and that a lack of vesicular zinc (ZnT3) is responsible for modifying the behavioral effects of social defeat stress [76].

Zinc deficiency has been shown to reduce the levels of doublecortin (DCX) expression, a marker of neuronal differentiation. Immunohistochemistry revealed a reduction in the number of cells that were double positive with BrdU and DCX [3,4]. In zinc-deficient mice, DCX-positive cells also reduced process formation and neuronal branching, which suggested that zinc deficiency actually impairs neurogenesis by reducing neural differentiation [3].

Extracellular matrix (ECM) adhesion and remodeling are critical for the cell migration of tissues [80]. In particular, migrating neuroblasts are known to play an important role in expressing matrix metalloproteinase (MMP). MMP is a zinc-dependent proteolytic enzyme that cleaves ECM components. MMP3 and MMP9 are expressed in neuroblasts in the rostral migratory stream (RMS) [81,82]. Blocking the expression or activity of these MMPs causes chemokine-induced migration in vitro, radial migration of the adult olfactory bulb, and striatal injury after stroke [83]. MMPs that are produced by activated endothelial cells are involved in the migration of neural cells after stroke [84]. In summary, neuroblasts can interact with the adult brain ECM, alter the ECM, and promote migration efficiency (Figure 2).

Therefore, zinc regulates multiple stages of neurogenesis, including cell proliferation, survival, migration, and differentiation.

### 2.3. Role of Zinc in Promoting Redox Homeostasis

Oxidative stress is an important factor in several chronic diseases that are associated with aging, such as atherosclerosis and related heart diseases, cancer, and neurodegeneration. O_2_^−^, H_2_O_2_, and OH are known as reactive oxygen species (ROS) and they are continuously produced in vivo under aerobic conditions [85,86].

Zinc is a well-known redox-inert metal. Redox-inert metals catalyze copper/zinc superoxide dismutase and they are involved in the protection of protein sulfhydryl groups and the upregulation of metallothionein (MT) expression. Taken together, zinc possesses a metal-binding capacity and exhibits antioxidant properties via several molecules and enzymes [87]. Nicotinamide adenine dinucleotide phosphate (NADPH) oxidase is a group of plasma membrane-associated enzymes that catalyze the production of O_2_^−^ from oxygen by using NADPH as an electron donor [88,89]. NADPH oxidase (Nox) activity is highly effective in suppressing superoxide release [90].

Zinc promotes the activation of molecules and enzymes, such as antioxidant proteins, glutathione (GSH), catalase, and SOD, reduces the activity of pro-oxidant enzymes, such as inducible nitrate synthase (iNos), and inhibits lipid peroxidation [91]. Zinc induces the generation of MT, a co-factor of the enzyme superoxide dismutase (SODs) that catalyzes the dismutation of O_2_^−^ to H_2_O_2_, which is a scavenger of OH [92,93,94,95,96].

Zinc regulates metalloenzymes by inhibiting activity in the absence of substrates or other weak chelators and it protects thiol groups from oxidation. Thiol (SH) groups have a particularly high affinity for the integrin IIB series metal ions Zn^2+^, Cd^2+^, and Hg^2+^. MT is a thiol-rich protein that binds preferentially to these ions and maintains high stability [97,98,99].

Thiol-containing compounds are central to various biological reactions. Disulfide bonds play an important role in determining the tertiary structure of proteins and, in many drugs, the cysteine moiety is an important reaction center that determines its effect. Molecules that contain cysteine residues are the most easily metabolized compounds and they are easily oxidized by transition metals or participate in thiol-disulfide exchange. GSH—which has one cysteine—and thioredoxin—which has two cysteines in its active site—are often complementary, if not overlapping, in cytoprotection. GSH is the most abundant non-protein thiol. Thioredoxin is a protein that contains a “thioredoxin fold” that plays an important role in antioxidant defense [100,101,102].

MT may play an important role in determining neuronal fate by regulating the localization and concentration of intracellular free zinc [103,104]. Recent studies have shown that oxidative and nitrosative stress during ischemic injury induced zinc accumulation and the activation of the signaling system associated with cell death processes. In particular, there is a link between the release of nitric oxide (NO)-derived species by intracellular zinc, and the role of this process and potassium channels in neuronal apoptosis. The chelation of intracellular zinc blocks the signal transduction pathway that is involved in the peroxynitrite-induced amplification of potassium currents [105].

Nuclear factor-erythroid 2-related factor 2 (Nrf2) is important in the prevention of oxidative damage. The transcription factor Nrf2 mediates adaptation to oxidative stress by inducing heme oxygenase-1 (HMOX1) and NAD(P)H Quinone Dehydrogenase 1 (NQO1), such as cytoprotective genes [106]. The zinc chelator N,N,N′,N′-Tetrakis(2-pyridylmethyl)ethylenediamine (TPEN) exacerbates oxidative damage that originally depleted intracellular free zinc and decreased Nrf2 expression and transcription [107]. Nrf2 is a member of the cap-n-collar/basic leucine zipper (CNC-bZIP) protein family and it is an important transcription factor that regulates the gene expression of antioxidant proteins and enzymes (GSH and SOD), detoxification enzymes (glutathione-s transferase-1 (GSTA-1), and hemeoxygenase-1 (HO-1), which binds to the antioxidant response element (ARE) in the promoter region of the target gene. Zinc can modulate the antiperoxidant properties that are effective in reversing oxidative stress that is induced by lithium toxicity in the brain [108]. Prasad et al. (1998) observed that zinc supplementation for healthy subjects aged 20–50 years reduced the concentration levels of MDA (malondialdehyde), 4-hydroxy alkenals (HAE), and 8-hydroxy deoxyguanine, which are oxidative stress markers in plasma [109].

Therefore, zinc has one of the strongest anti-oxidative profiles in the protection against oxidative stress.

### 2.4. Role of Zinc on Immunity

Zinc plays a central role in the immune system, and zinc-deficient patients show increased susceptibility to various pathogens. Zinc is essential for the normal development and function of cells that mediate nonspecific immunity, such as neutrophils and natural killer cells (NK cells). Specific cellular zinc transporters have been reported to be up and downregulated due to changes in increased zinc demand during inflammatory conditions [110]. Zinc deficiency has a dose-dependent response to plasma cytokines, as zinc prevalence affects the production of cytokines, such as interleukins (IL-1β, IL-2, and IL-6) and tumor necrosis factor alpha (TNF-α) by MT homeostasis, which is, in turn, affected by proinflammatory cytokines (Figure 3). The zinc deficiency model shows that, even in mild deficiencies, IL-1β is produced by monocytes, and this suggests that zinc deficiency activates monocytes and macrophages in order to produce inflammatory cytokines and increase oxidative stress itself. A statistical correlation between lymphocyte zinc and IL-1β generation has been shown [111,112]. NF-κB is an important transcription factor for the expression of inflammatory cytokines, which is reflected by extracellular and intracellular regulation. Zinc deficiency also affects the development of acquired immunity by preventing both T lymphocyte growth; zinc also has a specific activation functions for Th1 cytokine production and helps B-lymphocytes [113]. Zinc deficiency causes the atrophy of the thymus and lymphoid tissue in experimental animal models. It reduces the number of splenocytes and reduces the response to both T-cell-dependent and independent antigens [114].

Microglia are a specific class of immune cells in the central nervous system (CNS) [115,116]. When the CNS is challenged by infection, tissue injury, or other damage signals, it transforms into an activated form, which is termed the “ameboid”, and releases matrix metalloproteinases, reactive oxygen species (ROS), and other pro-inflammatory factors. Zinc is released by neurons under several conditions that cause microglial activation, and zinc chelation can reduce neuronal death in models of cerebral ischemia and neurodegenerative disorders. Kauppinen et al. (2008) showed that zinc directly triggers microglial activation. Zinc-induced microglia were transfected with a nuclear factor-κB (NF-κB) reporter gene, which showed a several-fold increase in NF-κB activity, which was blocked by inhibiting poly(ADP-ribose) polymerase-1 (PARP-1) and NADPH oxidase [117]. The connection between zinc and microglial activation represents a novel, unrecognized mechanism that may contribute to neuropathy.

The modulation of immune function by variation in zinc concentrations affects multiple aspects of physiology. Both extremes, either zinc deficiency and overloaded conditions, exacerbate brain damage or harmful immune response. Therefore, maintaining a moderate concentration of zinc is important in maintaining normal physiological function.

### 2.5. Role of Zinc in Synaptic Transmission

Synaptic zinc is present in presynaptic glutamatergic vesicles of a particular subset of glutamatergic synapses, called “zincergic” synapses, throughout multiple regions of brain, including the cerebral cortex, limbic system, hippocampus, and olfactory bulb (OB) [118].

It exhibits a role as a neuromodulator on a variety of membrane receptors, ion channels, and transporters [119]. Especially, synaptic zinc is enriched through a specific zinc transporter, ZnT3, and it is co-released with glutamate during exocytosis that is evoked by action potentials [120,121]. These co-released effluxes affect synaptic transmission to interact with receptors and channels to modulate auditory processing [122,123]. Synaptic zinc suppressed synaptic and extra-synaptic NMDA receptor (NMDAR), γ-aminobutyric acid types A (GABA-A), and calcium channels. Additionally, it potentiates α-amino-3-hydroxy-5-methyl-4-isoxazolepropionic acid (AMPA) and glycine amino acid receptor response [10,124,125,126,127]. Additionally, effects on other types of receptors, such as kainate receptors, serotonin receptors, dopamine receptors, acetylcholine receptors, and voltage-gated ion channels for Na^+^, K^+^, Ca^2+^, and Cl^−^ [124,128]. It is well known that zinc is available to modulate synaptic transmission that is mediated by excitatory (NMDA, AMPA) and inhibitory (GABA, glycine) receptors. The importance of NMDA receptors in the induction of long-term potentiation (LTP) and long-term depression (LTD) in the CA1 hippocampal region [129] and that zinc affects NMDA receptors as an endogenous antagonist [10].

Synaptic zinc regulates sensory processing and improves acuity in the discrimination of different sensory stimuli. Synaptic zinc plasticity causes long-term changes, which results in sensory experiences [130]. Recently, Vogler, N. W. et al. (2020) described the mechanism of this long-term synaptic zinc plasticity as being due to group 1 metabotropic glutamate receptors (G1 mGluRs) dependent mechanism that triggers a bidirectional long-term change in synaptic zinc signaling [130].

Therefore, synaptic zinc is able to modulate excitatory neurotransmission, which is mediated by excitatory and inhibitory receptors and controls sensory responses via G1 mGluR-dependent mechanisms.

## 3. Foe

The excitotoxic stimulation that occurs during ischemia or sustained status epilepticus releases a large amount of glutamate and zinc from presynaptic terminals, which leads to a large amount of zinc influx into postsynaptic neurons [123,131,132]. Increased intracellular free zinc that is driven by several deleterious stimuli causes extensive neuronal death [13].

One of the major mechanisms of zinc-induced neuronal cell death is known as oxidative stress. Previous studies have shown that zinc exposure significantly increases the levels of the NADPH oxidase subunit in both neurons and astrocytes [133,134]. NADPH oxidase is activated by protein kinase C (PKC), which is a key enzyme driving oxidative stress generation [135,136]. NADPH oxidase is a superoxide-generating enzyme that is composed of components located in the membrane (glycosylated 91-kDa glycoprotein (gp91PHOX) and p22PHOX) and located in the cytoplasm (p47PHOX, p40PHOX, and p67PHOX) [137,138] (Figure 4). The dynamic relationship between zinc and ROS is strongly associated with neuronal cell death in zinc-accumulation diseases, such as traumatic brain injury, ischemic stroke, and hypoglycemia [93].

### 3.1. Role of Zinc on Traumatic Brain Injury-Induced Neuronal Death

Traumatic brain injury (TBI) disturbs the normal structure and function of the brain due to a blow or jolt to the brain or a penetrating head injury. The initial stages of TBI development are associated with impaired cerebral blood flow and metabolism, reduced high-energy phosphate content, and oxygen consumption. The signs of TBI effects depend on the severity of the injury and may be mild, moderate, or severe, which range from ultrastructural damage to the mechanical destruction of large areas of the brain. The major source of damage is the result of mechanical insult to brain tissue from exposure to external forces, which includes axonal shear due to bruises, vascular injuries, and the stretching and tearing of nerve axons. Secondary injuries develop within minutes to months after the primary damage injury, leading to several deleterious cascades, such as metabolic, cellular, and molecular events that contribute to brain cell death, tissue damage, and atrophy [139]. Some reports have indicated that secondary injuries induce oxidative stress, and that this is a fundamental event that leads to the development and generalization of damage-generating mechanisms, such as glutamate toxicity, apoptosis, mitochondrial dysfunction, brain edema, autophagy, and inflammation [140,141].

Ischemia that is associated with TBI initiates a release of glutamate from presynaptic terminals after traumatic injury and leads to excitotoxicity and the cell death of postsynaptic neurons [142]. TBI is one of several brain diseases that are related to intracellular zinc. It is well established that TBI induces zinc accumulation in postsynaptic neurons, resulting in neuronal cell death [143,144]. Therefore, understanding zinc as an essential nutrient, but also as a potentially neurotoxic factor requires us to further examine the role of dietary zinc and zinc supplementation, particularly in neuropathy after stroke or TBI [144,145,146]. In the TBI-induced brain, zinc chelation by clioquinol (CQ) and TPEN substantially reduces the abnormal accumulation of intracellular zinc that contributes to brain damage through the promotion of neuronal apoptotic death, zinc translocation, excitotoxicity, and neural autophagy [147,148]. Previous reports have suggested that the modulation of zinc-overload after TBI has the potential for treating or preventing head trauma and its associated pathology.

Interestingly, TBI-induced zinc accumulation temporarily increases adult neurogenesis in the subgranular region (SGZ) of the hippocampus. Consequently, these phenomena can ultimately become drivers of neuronal death processes, which are associated with learning and memory deficits [147,149].

Dominguez et al. (2003) reported that zinc chelation contributes to nerve damage when the hippocampal neurons are exposed to overexcitation conditions that are induced by non-lesioning doses of kainic acid as a treatment plan in order to reduce the accumulation of zinc [150]. Hellmich HL et al. (2008) showed that treatment of extracellular zinc chelator, calcium salt of EDTA (CaEDTA), did not improve the Morris water maze performance after TBI-induced spatial learning and memory function deficits. However, CaEDTA reduced the number of damaged neurons in the hippocampal CA3 region. Additionally, they found that CaEDTA treatment increased the expression of the pro-apoptotic proteins BAX and caspase-3 two weeks after injury [151]. Doering et al. (2010) showed in subsequent studies that chelation or blocking of ionic zinc have adverse effects during acute phase in TBI. The chelation or block of ionic zinc with DEDTC or selenium increased both dead and apoptotic cells at 24 h post-TBI [152]. Li et al. (2010) reported that the chelation of cellular zinc ions after rapid stretch injury increases cellular ROS production [153].

On the other hand, evidence was recently presented that zinc chelation after brain injury could have adverse effects. Recent papers also suggested possible therapeutic effects of zinc deficiency in TBI, in contrast to zinc supplementation [154]. In particular, recent studies have suggested that zinc deficiency may be associated with alterations in MMP activity that may play a role in a variety of important functions that are associated with TBI, including the disruption of the blood-brain barrier, inflammation, and angiogenesis [155,156]. Additionally, zinc chelation has been reported to inhibit TBI induced neurogenesis [147].

In animal models, zinc supplementation may be an effective treatment option to improve behavioral outcomes, such as cognitive impairment and depression, following TBI in rat models [157]. Khazdouz M et al. (2018) recently confirmed the efficacy of zinc supplementation in patients with severe head trauma in a double-blind controlled study. In this trial, 100 patients with severe TBI were randomly assigned to a zinc supplement or a placebo group and then evaluated for 16 days. There was no difference between the two groups at baseline, but, interestingly, zinc supplementation groups showed significantly increased plasma zinc concentrations. For inflammation factors, the zinc supplementation group had significantly lower plasma c-reactive protein (CRP), erythrocyte sedimentation rate (ESR), and white blood cell (WBC) when compared to controls on day 16. At the same time point (day 16), the assessment of neurological recovery results, the zinc supplementation group had significantly lower sequential organ failure assessment (SOFA) scores than in the placebo group. The zinc supplementation group also had a significantly improved Glasgow Coma Scale (GCS) rating when compared with the placebo patient group [158]. Based on these recent studies, there is ample evidence to support zinc supplementation as having a potentially beneficial role for patients with conditions that are associated with an acute drop in plasma zinc levels, such as head trauma.

As mentioned above, zinc often has dual effects on cellular events; in the acute phase of head trauma, appropriately inhibition of zinc-overload can rescue brain impairment and, in the chronic phase, zinc supplementation can help to improve behavioral outcomes. Therefore, the regulation of zinc is important in solving TBI-induced cerebral, cognitive, and behavioral problems.

### 3.2. Role of Zinc on Hypoglycemia-Induced Neuron Death

Hypoglycemia in people with diabetes initially causes a brain fuel deficiency that triggers a series of physiological and behavioral defenses, but, if unchecked—usually after elevated plasma glucose levels—will result in functional brain failure being corrected. In rare cases, severe and prolonged hypoglycemia causes brain death [159,160,161]. The association between serum zinc levels and hypoglycemia has been previously studied. The serum zinc levels are directly correlated with glucose and lactate levels. These effects are secondary to zinc-mediated pancreatic alpha cell dysfunction, and hypoglycemia is impaired by the response of glucagon [162].

Suh et al. (2004) observed whether zinc release can mediate the poly (ADP-ribose) polymerase 1 (PARP-1) activation and neuronal death that results from severe hypoglycemia. The following four pieces of evidence in hypoglycemia-induced neurons were observed: the depletion of zinc from the presynaptic vesicles, the appearance of zinc in the somata of injured hippocampal postsynaptic neurons, the prevention of neuronal injury by the chelation of extracellular zinc, and the prevention of PARP activation by extracellular zinc chelation. Zinc can trigger PARP-1 activation, and PARP-1 activation can cause neuronal death [163] (Figure 5). PARP1 is a major DNA damage sensor whose (ADP-ribose) polymerase activity is rapidly regulated by interacting with DNA destruction. PARP1 recognizes DNA damage through the N-terminal DNA-binding domain that is composed of tandem repeats of the unusual zinc finger (ZnF) domains ZnF1 and ZnF2. The ZnF3 domain is the third zinc-binding domain of PARP-1; this is also essential for DNA-dependent PARP-1 activity [164,165].

The hypoglycemic brain experiences a transient increase in the number of proliferating progenitor cells and the number of immature neurons in rat SGZ after four weeks. Subsequently, a sustained decline of progenitor cell proliferation and immature neurons occurs [166]. Hypoglycemic neuronal cell death is not the simple result of a low glucose supply to the brain but is due to a cell death signaling pathway that is initiated by unregulated glucose reperfusion following a period of glucose deprivation. Excessive zinc release from presynaptic terminals and the subsequent translocation to postsynaptic neurons may lead to neuronal death after hypoglycemia. Glucose reperfusion causes hypoglycemic neuronal death. Specifically, the primary source of superoxide generation in hypoglycemia-reperfusion is neuronal Nox. Hypoglycemia-induced neuronal Nox activation involves zinc as a primary signaling molecule [167].

Kho et al. (2017) suggested that N-Acetyl-L-cysteine (NAC) acts as a zinc chelator that can alleviate the zinc-induced neuronal cell death process [168]. Additionally, NAC restores neuronal GSH levels, which have a potent role in antioxidant protection by providing a cell-permeable cysteine source. In addition to increasing GSH levels, NAC treatment can also reduce neuronal cell death by zinc chelation. NAC reduces oxidative stress, zinc release, and translocation, and it improves glutathione levels in neurons. Thus, NAC treatment attenuates hippocampal neuronal death in hypoglycemic-induced rats [168].

Blood glucose deficiency-induced hypoglycemic brain damage can promote the release of vesicular zinc by nitric oxide (NO) synthesis and contributes to an acceleration of hypoglycemic neuronal damage. From this point of view, the regulation of zinc translocation by hypoglycemia is an example of a therapeutic target for decreasing neuronal death by acute zinc chelation.

### 3.3. Role of Zinc on Ischemia-Induced Neuronal Death

Zinc can function as both a signal transduction mediator and a neurotoxin and it has been implicated in cerebral ischemia. Zinc exerts neuroprotective or neurotoxic effects during global and focal cerebral ischemia [11,169,170,171]. Vesicular glutamate release is aggravated by ischemic stroke, and protocols for detection by microdialysis confirm cerebral ischemia-induced extracellular zinc accumulation [172,173]. Ischemic stroke-induced excessive zinc release from synaptic vesicles triggers excitotoxicity that moves into post-synaptic neurons through calcium-permeable channels, such as GluR and calcium-permeable AMPA/K (Ca-A/K). In experimental global ischemia conditions, high concentrations of chelatable zinc have been known to be a critical mediator of the neuronal degeneration processes [174,175]. During ischemia, high concentrations of zinc released from a subset of glutamatergic terminals may promote zinc translocation and accumulation in vulnerable postsynaptic neurons. Following this logic, modulating zinc release and controlling concentrations within a physiological range may be of clinical importance. A further hierarchy of susceptibility to ischemic injury among various types of neurons exists. Hippocampal cornus ammonis 1 (CA1) pyramidal cells and cerebellar Purkinje cells appear to be the most vulnerable neurons in the brain in the context of several neurological diseases, followed by striatal medium-sized neurons, and neocortical neurons in layers 3, 5, and 6 [176].

ZnT-1 (zinc transporters-1) and metallothioniene III can be upregulated to promote zinc efflux and cytoplasmic buffering, respectively, in order to regulate the ionic gradients of cytoplasmic zinc levels during cerebral ischemic conditions. Therefore, the induction of ZnT-1 mRNA expression was upregulated in the CA1 subfield of the hippocampus after global ischemia. In response to increased levels of intracellular zinc, we hypothesize that vulnerable neurons experience an upregulated expression of ZnT-1, a plasma membrane-localized zinc transporter that promotes zinc efflux in order to maintain zinc homeostasis [11]. ZnT-1 is highly expressed at the hippocampal postsynaptic density (PSD), where NMDA receptors (NMDARs) are enriched, and it specifically binds the GluN2A subunit of the NMDARs. The ZnT-1/GluN2A complex is regulated by the induction of synaptic plasticity at the glutamatergic synapse [177]. It suggests that synaptic zinc inhibition of NMDARs required postsynaptic intracellular zinc; also, it suggests that cytoplasmic zinc is transported by ZnT-1 to the extracellular space in close proximity to the NMDAR [178].

Zinc also acts in an essential role of promoting calcium-permeable AMPA/kainite (Ca-A/K) channel subunit expression during ischemia by altering transcriptional regulation, leading to the downregulation of the GluR2 subunit that renders the A/K receptor channel calcium-impermeable [179]. It was hypothesized that zinc-related reduction in GluR2 expression may allow for the entry of toxic calcium and even zinc during ischemia [180,181]. Furthermore, in the ischemic brain, increased cytosolic zinc levels have been shown to induce the expression of the zinc finger transcription factor REST (restrictive element 1 silencing transcription factor), which is able to repress neuro-specific target genes, including GluR2 [182,183]. Zinc also potentiates glutamate-induced nerve injury by directly inhibiting GABAA channels and blocking glutamate reuptake by blocking the excitatory amino acid transporter (EAAT-1) that is expressed in glial cells. The intraperitoneal pretreatment of zinc alone or in combination with a GABAA antagonist—bicuculline—had a detrimental effect on the development of an infarct after the induction of neurological deficits and focal ischemia [184].

In ischemic stroke, two sources of ROS production exist, including the production of mitochondrial ROS by the release of zinc. Interestingly, in the chronic phase, zinc accumulation activates neuronal NADPH oxidases. ROS that is produced by NADPH oxidase further increases zinc accumulation, leading to cell death and ischemic brain injury [185,186] (Figure 4 and Figure 6).

Zinc overload during cerebral ischemia can trigger cell death pathways that involve ROS production through mitochondrial dysfunction and the activation of NADPH oxidase. Therefore, inhibition of ischemic stroke-induced excessive vesicular zinc releasing may contribute to attenuation of ischemic pathology.

### 3.4. Role of Zinc on Alzheimer’s Disease

Alzheimer’s disease (AD) is a progressive neurodegenerative disease that symptomatically shows progressive cognitive decline, such as memory loss, speech problems, learning and orientation difficulties, and behavioral changes [187]. It also causes non-memory linked behavioral symptoms, such as depression, anxiety, and agitation. Pathologically, AD is characterized by amyloid deposits that consist of extracellular aggregated amyloid beta protein (Aβ) and intracellular neurofibrillary tangles (NFT). The amyloid precursor protein (APP) and Aβ have zinc-binding sites [188]. Zinc accentuates Aβ toxicity, and zinc sequestration into amyloid deposits leads to the loss of functional zinc at the synapse [189]. Many reports have suggested that zinc transporters are involved in the formation of AD senile plaques. The enriched expression and modified distribution of zinc transporters have been found in senile plaques in the human AD brain and amyloid precursor protein (APP) or APP/presenyllin (PS) mouse models. The missense mutations in the APP, PS1, and PS2 genes are common pathogenic mechanisms that cause abnormalities in APP metabolic processes and induce the accumulation of Aβ [188,190,191,192].

Surprisingly, the expression levels of the zinc transporter are altered abnormally during AD progression, including six ZnT transporter families and one Zrt-/Irt-like protein (ZIP) transporter. Some alterations of ZnT proteins (ZnT-1, ZnT-4, and ZnT-6) were found in the progression of AD [193]. ZnT1 and ZnT4 were present in the entire senile plaque, ZnT3, ZnT5, and ZnT6, were present in the periphery of the plaque, and ZnT7 was expressed in the center of the senile plaque [194]. Specifically, ZnT3 levels were further decreased in the AD cortex. Thus, synaptic zinc release may be decreased in AD. Adlard et al. (200) showed that ZnT3 knockout mice may model an important aspect of AD synaptic pathology [195]. The loss of trans-synaptic zinc movement leads to cognitive loss and, since extracellular Aβ is aggregated by and sequesters zinc, the genetic ablation of ZnT3 shows a phenomenon for the synaptic and memory deficits of AD.

The NMDA receptor contributes to zinc toxicity by providing a route of the zinc influx pathway into neurons. In other words, the mis-compartmentalization of zinc in AD interferes with signaling by the NMDA receptor, which can lead to intracellular excitatory toxicity [196] (Figure 7). Additionally, zinc deficiency (or low levels of zinc) is commonly found in AD patients, and most cases show that it worsens AD-induced memorial deficits and behavioral outcomes. In mouse models, it has been shown that a zinc-deficient diet causes memory decline. Zinc deficiency is also suggested to cause changes in behavior and mood, including depression and anxiety [197].

Aβ oligomers (Aβ) interact with intracellular zinc and lead to decreased levels of labile zinc [198]. Additionally, the Aβ induced ZnR/GPR39-dependent Ca^2+^ response was significantly reduced, and it resulted in much lower activation of MAPK [199]. Decreased ZnR/GPR39-dependent signaling reduces KCC2 activation. This may contribute to an increase in seizures found in Alzheimer’s disease patients [44].

Therefore, abnormal homeostasis of zinc may have a critical role in the AD, and identifying a suitable agent for zinc chelation in this setting may be a potential therapeutic target.

## 4. Conclusions

Zinc is a trace metal ion in the central nervous system that plays three major biological roles in catalysis, structure, and regulation. Zinc contributes to antioxidant function and the proper functioning of the immune system and it plays an important role in neurophysiology, which ranges from cell growth to cell proliferation. It also plays a critical pathophysiological role in major neurological diseases. Zinc homeostasis is important, representing a point of intersection in related brain diseases. In particular, excessively released and accumulated zinc ions after brain insults cause neurotoxic damage to postsynaptic neurons. On the other hand, zinc deficiency causes neurodegenerative and cognitive decline disorders, such as decreased neurogenesis and impaired learning and memory.

For these reasons, the concept of preventive treatment, in which a disease that is related to zinc accumulation is prevented by reducing the level of zinc through a metal-chelator, is a promising method that needs further exploration. Zinc-chelating agents can bind to metal ions and form complex structures that are easily excreted by the body to remove them from the intracellular or extracellular spaces. However, these metal chelates have side effects not only on zinc concentrations, but also on other important metals [200,201,202]. In addition, zinc deficiency could be associated with DNA fragmentation, and increased cell death and apoptosis at the site of injury after head trauma. Additionally, it impairs cell proliferation and survival through several signaling pathways (such as p53) in the brain [2,203,204,205].

Conversely, in the case of degenerative disorders, due to zinc deficiency, such as TBI, zinc supplementation through dietary intake may be a beneficial therapeutic concept. In light of therapeutic zinc supplementation, it improved the rate of neurologic recovery and visceral protein concentrations and in reducing mortality and GCS recovery in patients with severe brain injury [206]. However, it is necessary to assess the optimal response by determining the effective dosage, timing, and therapeutic outcomes for zinc treatment in head injury patients. The administration of zinc can potentially interfere with copper absorption and it can cause anemia, neutropenia, and copper deficiency that are associated with neurological symptoms [207,208,209].

Therefore, a very important aspect of understanding how zinc can be used as a therapeutic agent is the necessity of a detailed understanding of the factors that influence zinc ion homeostasis and the optimization of the relevant concentrations.

## Figures and Tables

**Figure 1 ijms-21-08941-f001:**
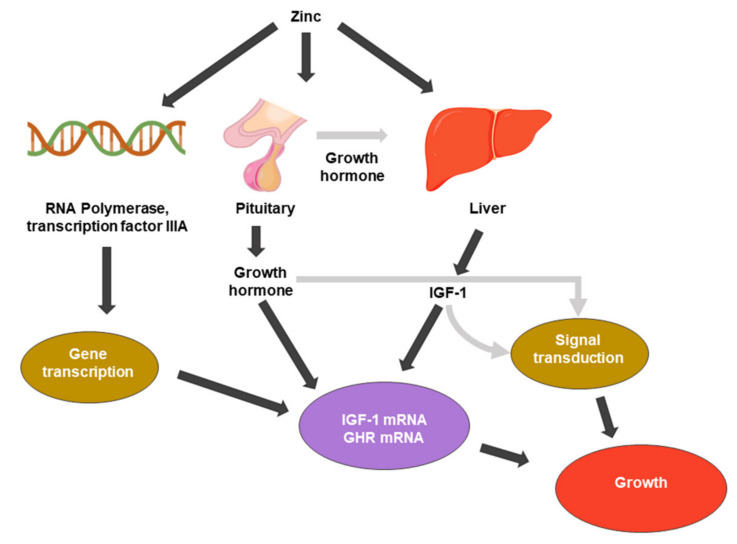
Zinc metabolic processes associated with cell growth. IGF-1: insulin-like growth factor 1. GHR: growth hormone receptor.

**Figure 2 ijms-21-08941-f002:**
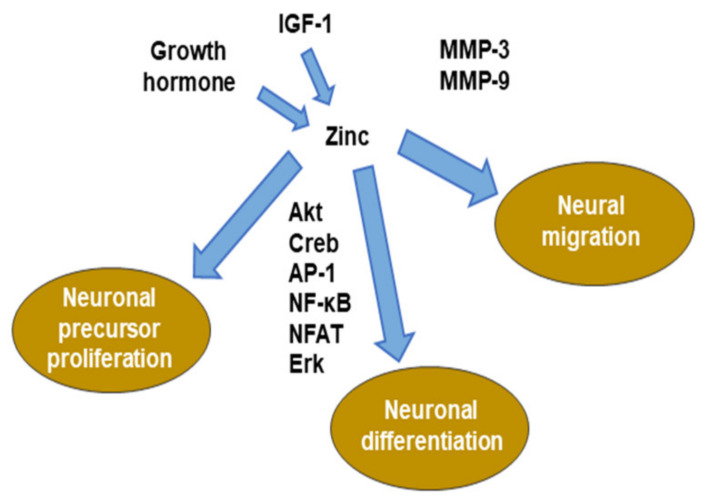
Role of zinc in neurogenesis. MMP: matrix metalloproteinase. ERK: extracellular signal-regulated kinase. NFAT: nuclear factor of activated T-cells. AP-1: activator protein-1.

**Figure 3 ijms-21-08941-f003:**
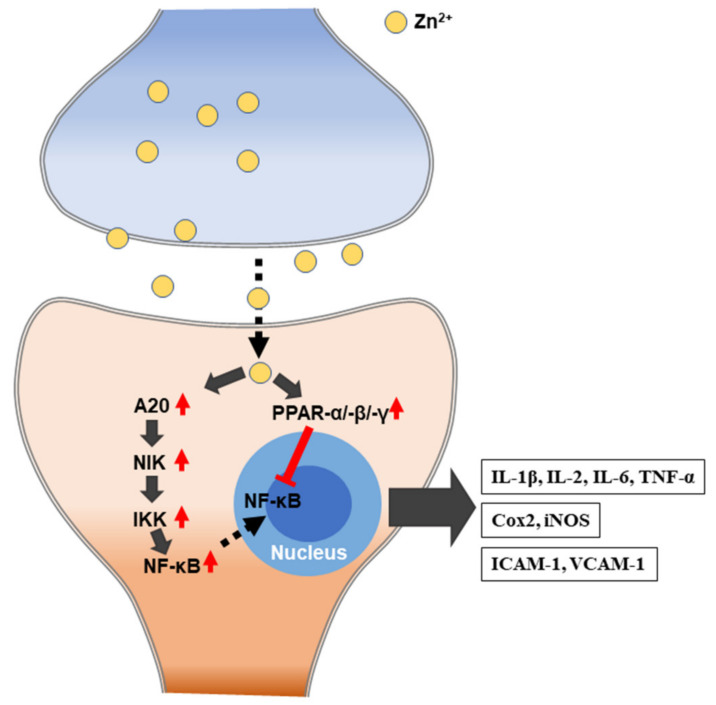
Schematic illustration of zinc-induced immune responses. IL: interleukin; TNF-α: tumor necrosis factor alpha. Cox2: cyclooxygenase2. iNOS: inducible nitric oxide synthase. ICAM-1: intercellular adhesion molecule 1. VCAM-1: vascular cell adhesion protein 1.

**Figure 4 ijms-21-08941-f004:**
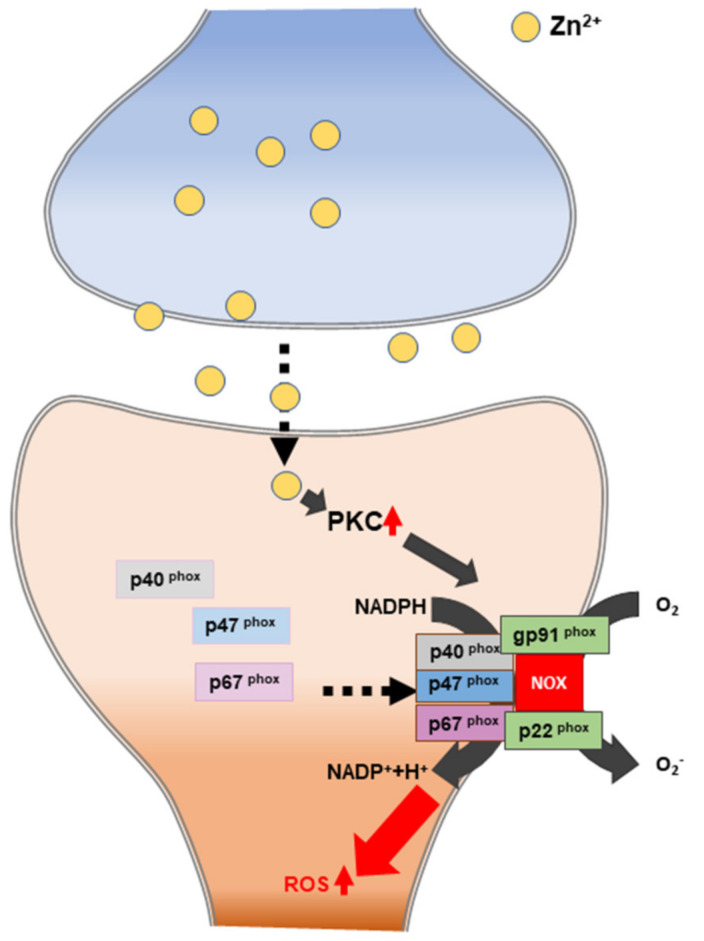
Schematic illustration of zinc-induced reactive oxygen species (ROS) generation. PKC: protein kinase C. NADPH: nicotinamide-adenine dinucleotide phosphate.

**Figure 5 ijms-21-08941-f005:**
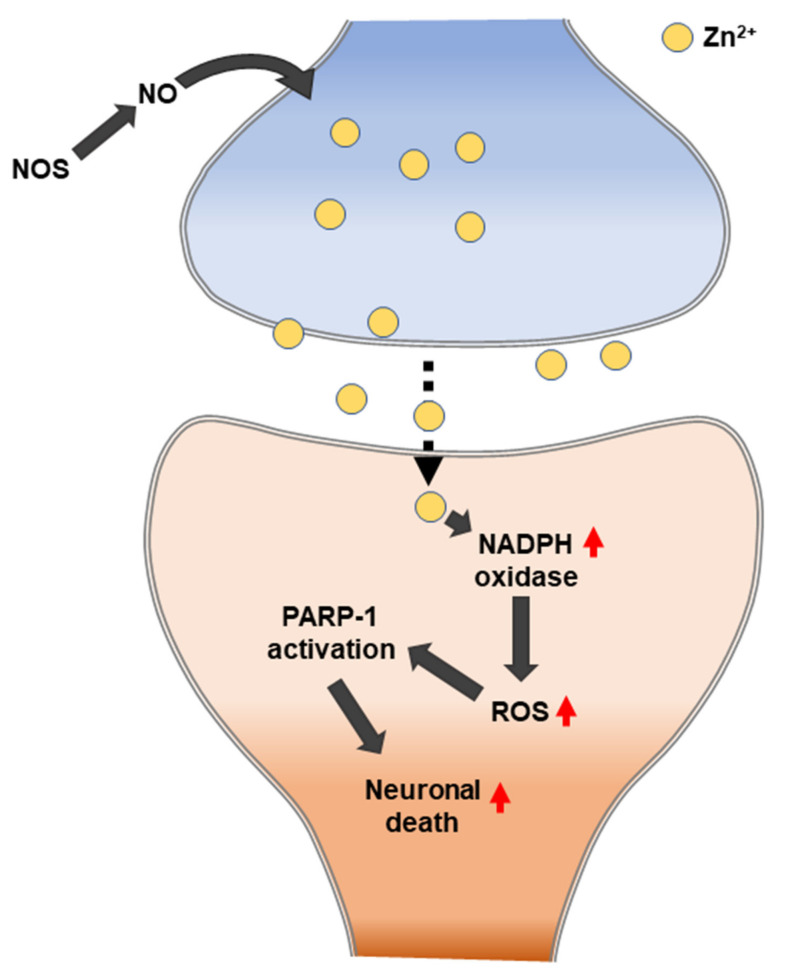
Hypoglycemia-induced neuronal death caused by zinc translocation.

**Figure 6 ijms-21-08941-f006:**
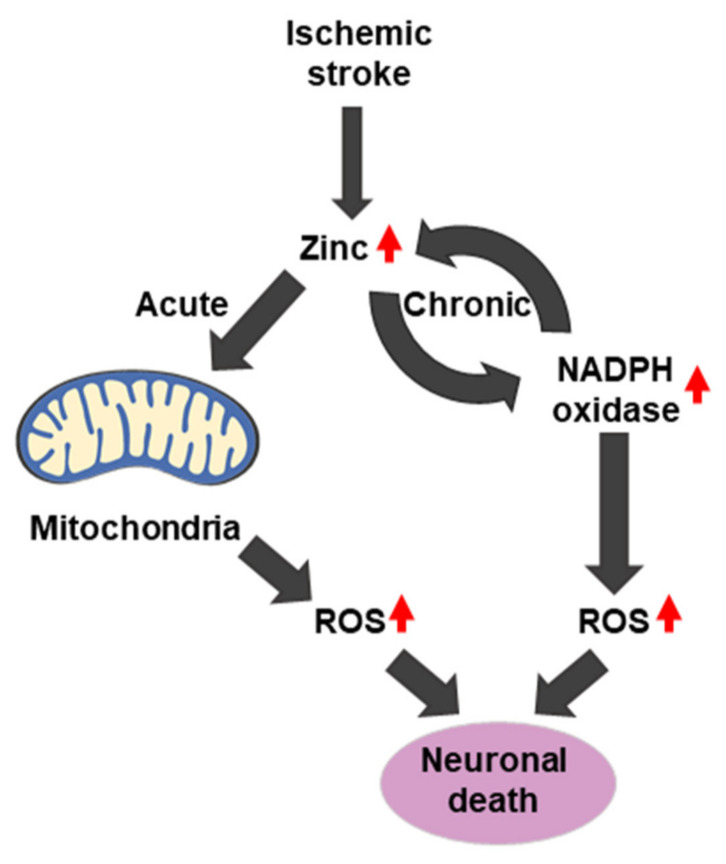
Zinc and ROS interaction in ischemic stroke.

**Figure 7 ijms-21-08941-f007:**
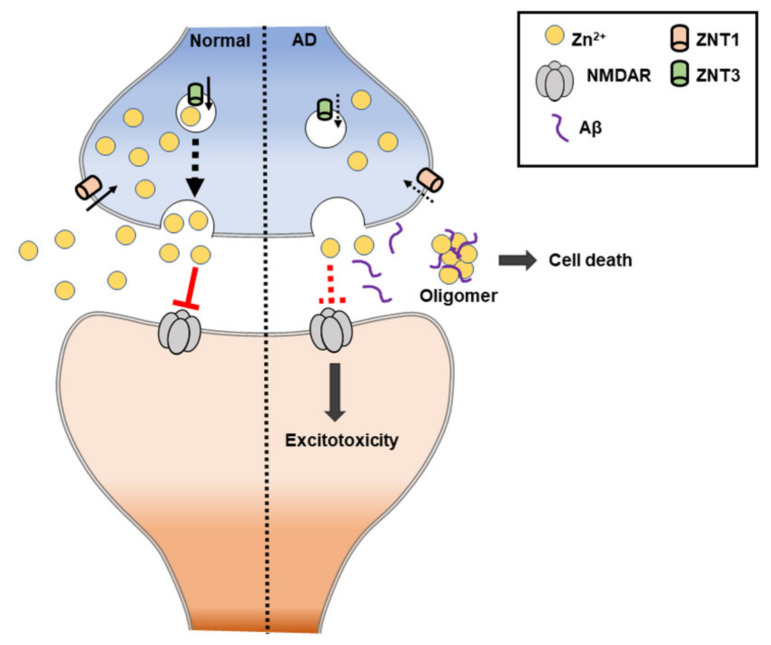
Schematic illustration of zinc-induced Alzheimer’s disease (AD). ZnT: zinc transporter. NMDAR: N-methyl-D-aspartate receptor.

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
