# Peer review of "Zinc in the Brain: Friend or Foe?"

_ijms, 2020, doi:10.3390/ijms21238941_

Round 1

Reviewer 1 Report

This review addresses the continuing controversy about the role of zinc in the brain. This issue is particularly important under conditions of neuronal damage from seizure, stroke, TBI, and other insults.

This review is strengthened by its focus on mechanisms of zinc action, many of which have been ignored in previous reviews on this topic. 

One area that should be addressed more thoroughly is the role of zinc in neuronal damage. in line 304 the authors report that, “Previous reports have suggested that the modulation of zinc-overload after TBI has potential for treating or preventing head trauma and its associated pathology”. While there are data, cited in the manuscript, that chelation prevents zinc accumulation after injury, there is a significant amount of work suggesting that this is not neuroprotective in vivo. In fact, there is work (not cited in this manuscript) showing that chelation of zinc causes neuronal damage after TBI and other forms of injury.

  • Dominguez et al., 2003
  • Hellmich et al., 2008
  • Doering et al., 2010
  • Li et al., 2010

Another set of work that sheds light on the role of zinc after neuronal damage is in vivo supplementation data. There are several papers that address this issue that are not cited by this  work, and would likely alter the overall conclusions about the role of zinc. This includes human work as well as pre-clinical animal work.

  • Young et al., 1996
  • Cope et al., 2011
  • Cope et al., 2012
  • Cope et al., 2016
  • Khazdouz, et al., 2018

Author Response

Dear Dr. Susana Solá and Sara Xapelli, Guest Editor, International Journal of Molecular Sciences

I appreciate the opportunity to revise this manuscript. I respect the editor’s and reviewer’s helpful comments and have responded to both reviewer comments point-by-point, revised as indicated below <yellow highlights>. I hope this revised manuscript is acceptable for publication in your journal.

Reviewer #1: This review addresses the continuing controversy about the role of zinc in the brain. This issue is particularly important under conditions of neuronal damage from seizure, stroke, TBI, and other insults.

This review is strengthened by its focus on mechanisms of zinc action, many of which have been ignored in previous reviews on this topic.

One area that should be addressed more thoroughly is the role of zinc in neuronal damage. in line 304 the authors report that, “Previous reports have suggested that the modulation of zinc-overload after TBI has potential for treating or preventing head trauma and its associated pathology”. While there are data, cited in the manuscript, that chelation prevents zinc accumulation after injury, there is a significant amount of work suggesting that this is not neuroprotective in vivo. In fact, there is work (not cited in this manuscript) showing that chelation of zinc causes neuronal damage after TBI and other forms of injury.

Dominguez et al., 2003

Hellmich et al., 2008

Doering et al., 2010

Li et al., 2010

<Response: We appreciate your helpful comments. In section 3.1. role of zinc on traumatic brain injury-induced neuronal death on page 10, we added an additional paragraph addressing the reviewer’s suggestion as follows.

“Dominguez et al. (2003) reported that zinc chelation contributes to nerve damage when the hippocampal neurons are exposed to overexcitation conditions induced by non-lesioning doses of kainic acid as a treatment plan to reduce the accumulation of zinc [150]. Hellmich HL et al. (2008) showed that treatment of extracellular zinc chelator, calcium salt of EDTA (CaEDTA) did not improve the Morris water maze performance after TBI-induced spatial learning and memory function deficits. But, CaEDTA reduced the number of damaged neurons in the hippocampal CA3 region. Additionally, they founded that CaEDTA treatment increased expression of the pro-apoptotic proteins BAX and caspase-3 two weeks after injury [151]. Doering et al. (2010) showed subsequent studies, chelation or blocking of ionic zinc have adverse effects during acute phase in TBI. Chelation or block of ionic zinc with DEDTC or selenium, increased both dead and apoptotic cells at 24 hours post-TBI [152]. Li et al. (2010) reported that chelation of cellular zinc ions after rapid stretch injury increases cellular ROS production [153]”>

Another set of work that sheds light on the role of zinc after neuronal damage is in vivo supplementation data. There are several papers that address this issue that are not cited by this work, and would likely alter the overall conclusions about the role of zinc. This includes human work as well as pre-clinical animal work.

Young et al., 1996

Cope et al., 2011

Cope et al., 2012

Cope et al., 2016

Khazdouz, et al., 2018

<Response: We appreciate your meaningful comments. In the Conclusion section on page 15, we added several studies that the reviewers mentioned as follows.

“In light of therapeutic zinc supplementation, Young et al. (1996) showed that zinc supplementation improved the rate of neurologic recovery and visceral protein concentrations and in reducing mortality and Glasgow Coma Scale (GCS) recovery in patients with severe brain injury [195]. In animal models, zinc supplementation may be an effective treatment option to improve behavioral outcomes such as cognitive impairment and depression following TBI in rat models [196]. However, it necessary to assess the optimal response by determining the effective dosage, timing, and therapeutic outcomes for zinc treatment in head injury patients. In addition, mild zinc deficiency could be associated with DNA fragmentation, and increased cell death and apoptotic at the site of injury after head trauma. Zinc deficiency also impairs cell proliferation and survival through several signaling pathways (such as p53) in the brain [2,197-199].”

Reviewer 2 Report

This is a nice review, presented in a provocative and interesting manner egarding the double edge sword effects of zinc. Unfortunately, it lacks references to many of the more recent studies in the field, and focuses mostly on earlier studies that have been discussed in numerous reviews. I strongly urge the authors to update the review and make it more appealing as an update on the revolutions that occurred in the field (focusing on neuronal aspects and recent studies if needed to shorten the ref list). Thus to make this review relevant for publication, more recent studies need to be discussed in more detail, important examples are suggested below:

The studies of Dyck, describing neuronal morphology and neurogenesis should be addressed and associated with the behavioral phenotypes. Specific studies using the ZnT3 Ko mice are of relevance and should be discussed in relation to the authors results using these mice (e.g. on page 4).

The role of ZnR/GPR39 in both cell growth in general and particularly neuronal function, with an important regulation of neuronal excitability via regulation of KCCs should be described. Moreover, this has important impact on neuronal inhibitory tone and seizure. In addition, a recent study described the role of this receptor in regulation of MMP in breast cancer cells and this could add a missing link of zinc regulation to the discussion on this pathway (i.e. page 5). Finally, a link between this receptor and impaired signaling during AD has been described, and could add to the discussion on page 11.

Recent important studies by the group of Tzounopoulos on the role of zinc in modulating synaptic strength, spontaneous firing and underlying regulation of NMDA, AMPA and glycine channels, and the resulting effects on frequency discrimination show a clear role for zinc in sensory function and should be discussed.

The role of zinc released from metallothioneins during oxidative and nitorsative stress and the seminal studies of Aizenman and Rosenberg on neuronal toxicity are missing in this review. More recent studies on its regulation of potassium channels, are hallmark studies which address the mechanism of zinc toxicity absent in the earlier works that are cited in this review.

A discussion on ZnT-1 on page 10 should relate to the ZnT-1 link with NMDA receptor signaling (Mellone, J Neurochem. 2015; and Krall, Sci Adv. 2020).

Technical:

References are sometimes mentioned with the name of the author in the text, but the number within the references list is missing, see for example line 131, page 4 or line 342 on page 10.

Note that 2+ should always be superscript.

Sentences in section 3.1, 3.2, 3.3 and others should be combined into a paragraph.

Ref to figures should be corrected, for example, page 11 is likely not Fig. 2.

Author Response

Dear Dr. Susana Solá and Sara Xapelli, Guest Editor, International Journal of Molecular Sciences

I appreciate the opportunity to revise this manuscript. I respect the editor’s and reviewer’s helpful comments and have responded to both reviewer comments point-by-point, revised as indicated below <yellow highlights>. I hope this revised manuscript is acceptable for publication in your journal.

Reviewer #2: This is a nice review, presented in a provocative and interesting manner egarding the double edge sword effects of zinc. Unfortunately, it lacks references to many of the more recent studies in the field, and focuses mostly on earlier studies that have been discussed in numerous reviews. I strongly urge the authors to update the review and make it more appealing as an update on the revolutions that occurred in the field (focusing on neuronal aspects and recent studies if needed to shorten the ref list). Thus to make this review relevant for publication, more recent studies need to be discussed in more detail, important examples are suggested below:

The studies of Dyck, describing neuronal morphology and neurogenesis should be addressed and associated with the behavioral phenotypes. Specific studies using the ZnT3 Ko mice are of relevance and should be discussed in relation to the authors results using these mice (e.g. on page 4).

<Response: We appreciate your helpful comments. We added additional cases in section 2.2. Zinc increases neurogenesis on page 5 as follows.

“ZnT3 knockout mice showed increased basal dendrite length of layer 2/3 pyramidal neurons of the barrel cortex, compared to wild-type mice [71].

As numerous previous studies have described, ZnT3 is involved in the regulation of hippocampal neurogenesis. In particular, ZnT3 KO mice show abnormalities in adult hippocampal neurogenesis compared with WT mice in response to hypoglycemia [4]. Some research groups have performed interesting studies on how neurogenesis is regulated by environment and stress. In particular, neurogenesis can be enhanced in rodents by transferring environments (from standard laboratory housing to more complex, enriched environments) and also by application of serotonin reuptake inhibitors (SSRI) [72,73]. However, ZnT3 KO mice do not show improved neurogenesis in response to SSRI treatment [74] or the benefits of enriched housing [75].

McAllister et al., (2020) confirmed that ZnT3 KO mice under RSD stress may regulate neurogenesis through an increased survival rate of newly born cell, similar to WT mice under RSD [76]. However, there is still no evidence of changes in ZnT3 behavior by fluoxetine treatment. Therefore, it partially supports the previous findings [77-79] that vesicular zinc is required for the neurogenic effects of chronic fluoxetine treatment and that lack of vesicular zinc (ZnT3) is responsible for modifying the behavioral effects of social defeat stress [76].”>

The role of ZnR/GPR39 in both cell growth in general and particularly neuronal function, with an important regulation of neuronal excitability via regulation of KCCs should be described. Moreover, this has important impact on neuronal inhibitory tone and seizure. In addition, a recent study described the role of this receptor in regulation of MMP in breast cancer cells and this could add a missing link of zinc regulation to the discussion on this pathway (i.e. page 5). Finally, a link between this receptor and impaired signaling during AD has been described, and could add to the discussion on page 11.

<Response: We appreciate your helpful comments. We added additional text on pages 3 and 13. section 2.1. Zinc is essential for cell growth and 3.4. Role of zinc on Alzheimer's diseases in the revised manuscript.

“The Zn2+-sensing G-protein coupled receptor (ZnR/GPR39) has been shown to regulate the activity of ion transport mechanisms that are important for the physiological function of epithelial and neuronal cells [37]. In particular, ZnR/GPR39 activity enhanced neuronal inhibitory tone. Zinc deficiency has been linked to epilepsy and seizures, suggesting an important physiological role for ZnR/GPR39 [38-40]. This specific extracellular receptor has been strongly linked to regulation of the mitogen-activated protein kinase (MAPK) and PI3K pathways [41-44]. Additionally, it triggers the activation of ERK1/2, which, in turn, activates p21 and cyclin D1 [45]. Activation of ZnR/GPR39 by Zn2+ also upregulates K+/Cl co-transport (KCC) activity. These ZnR/GPR39 and KCC inputs are essential for mediating Zn2+-dependent cellular signaling pathways leading to cell survival, migration and proliferation [46]

“Aβ oligomers (Aβ) interact with intracellular zinc and lead to decreased levels of labile zinc [193]. Also, the Aβ induced ZnR/GPR39-dependent Ca2+ response was significantly reduced and it resulted in much lower activation of MAPK [194]. Decreased ZnR/GPR39-dependent signaling reduces KCC2 activation. This may contribute to increase seizures found in Alzheimer’s disease patients [44].>

Recent important studies by the group of Tzounopoulos on the role of zinc in modulating synaptic strength, spontaneous firing and underlying regulation of NMDA, AMPA and glycine channels, and the resulting effects on frequency discrimination show a clear role for zinc in sensory function and should be discussed.

<Response: Thank you for the reviewer’s comments. We wrote on page 8, section 2.5. Role of zinc in synaptic transmission in revised manuscript.

“2.5. Role of zinc in synaptic transmission

Synaptic zinc is present in presynaptic glutamatergic vesicles of a particular subset of glutamatergic synapses, called "zincergic" synapses, throughout multiple regions of brain, including the cerebral cortex, limbic system, hippocampus and olfactory bulb (OB) [118].

It exhibits a role as a neuromodulator on a variety of membrane receptors, ion channels and transporters [119]. Especially, synaptic zinc is enriched through a specific zinc transporter, ZnT3, and it is co-released with glutamate during exocytosis evoked by action potentials [120,121]. These co-released effluxes affect synaptic transmission to interact with receptors and channels to modulate auditory processing [122,123]. Synaptic zinc suppressed synaptic and extra-synaptic NMDA receptor (NMDAR), γ-aminobutyric acid types A (GABA-A) and calcium channels. Additionally, it potentiates α-amino-3-hydroxy-5-methyl-4-isoxazolepropionic acid (AMPA) and glycine amino acid receptor response [10,124-127]. And effects on other types of receptors such as kainate receptors, serotonin receptors, dopamine receptors, acetylcholine receptors, and voltage-gated ion channels for Na+, K+, Ca2+, and Cl [124,128]. It is well known that zinc is available to modulate synaptic transmission mediated by excitatory (NMDA, AMPA) and inhibitory (GABA, glycine) receptors. The importance of NMDA receptors in the induction of long-term potentiation (LTP) and long-term depression (LTD) in the CA1 hippocampal region [129] and that zinc affects NMDA receptors as an endogenous antagonist [10].

Synaptic zinc regulates sensory processing and improves acuity in discrimination of different sensory stimuli. Synaptic zinc plasticity causes long-term changes, resulting in sensory experiences [130]. Recently, Vogler, N. W. et al (2020) described the mechanism of this long-term synaptic zinc plasticity as being due to group 1 metabotropic glutamate receptors (G1 mGluRs) dependent mechanism that triggers a bidirectional long-term change in synaptic zinc signaling [130]”>

The role of zinc released from metallothioneins during oxidative and nitorsative stress and the seminal studies of Aizenman and Rosenberg on neuronal toxicity are missing in this review. More recent studies on its regulation of potassium channels, are hallmark studies which address the mechanism of zinc toxicity absent in the earlier works that are cited in this review.

<Response: We appreciate your helpful comments. We added additional paragraph in “2.3. Role of zinc in promoting redox homeostasis” section in the revised manuscript.

“MT may play an important role in determining neuronal fate by regulating the localization and concentration of intracellular free zinc [103,104]. Recent studies have shown that, oxidative and nitrosative stress during ischemic injury induced zinc accumulation and activation of the signaling system associated with cell death processes. In particular, there is a link between the release of nitric oxide (NO)-derived species by intracellular zinc, and the role of this process and potassium channels in neuronal apoptosis. Chelation of intracellular zinc blocks the signal transduction pathway involved in the peroxynitrite-induced amplification of potassium currents [105]”>

A discussion on ZnT-1 on page 10 should relate to the ZnT-1 link with NMDA receptor signaling (Mellone, J Neurochem. 2015; and Krall, Sci Adv. 2020).

<Response: We appreciate your helpful comments. We added an additional paragraph on page 12 3.3. Role of zinc on ischemia-induced neuronal death section in revised manuscript.

“ZnT-1 is highly expressed at the hippocampal postsynaptic density (PSD) where NMDA receptors (NMDARs) are enriched and it specifically binds the GluN2A subunit of the NMDARs. ZnT-1/GluN2A complex is regulated by induction of synaptic plasticity at the glutamatergic synapse [172]. It suggests that synaptic zinc inhibition of NMDARs required postsynaptic intracellular zinc, also it suggests that cytoplasmic zinc is transported by ZnT-1 to the extracellular space in close proximity to the NMDAR [173].

References are sometimes mentioned with the name of the author in the text, but the number within the references list is missing, see for example line 131, page 4 or line 342 on page 10.

<Response: We re-wrote it in revised manuscript.>

Note that 2+ should always be superscript.

<Response: We re-wrote it in revised manuscript.>

Sentences in section 3.1, 3.2, 3.3 and others should be combined into a paragraph.

<Response: We corrected and re-wrote it in revised manuscript.>

Ref to figures should be corrected, for example, page 11 is likely not Fig. 2.

<Response: We modified it in revised manuscript.>

Round 2

Reviewer 1 Report

This review is has been significantly improved by the addition of relevant and important literature. 

The only remaining issue is the conclusion. The revised conclusion appears to now include new papers not discussed elsewhere in the manuscript. There are several reasons why this is problematic. First, the conclusion should not present new information, it should articulate a final conclusion based on the analysis of the totality of the data analyzed in the review. Secondly, the work that appears in the conclusion of the paper is not fully explained or evaluated. There is simply not enough information about the the use of supplementation. It is important add more information, particularly given that the most recent study (Khazdouz et al., 2018) is both a human clinical trial and to my knowledge does not appear in any of the other numerous reviews on this topic. Thus, the data in the newly cited papers need to be discussed in more depth in the body of the paper. 

Author Response

Dear Dr. Susana Solá and Sara Xapelli, Guest Editor, International Journal of Molecular Sciences

I appreciate the opportunity to revise this manuscript. I respect the editor’s and reviewer’s helpful comments and have responded to the reviewer comments point-by-point, revised as indicated below <yellow highlights>. I hope this revised manuscript is acceptable for publication in your journal.

Reviewer #1: This review is has been significantly improved by the addition of relevant and important literature.

The only remaining issue is the conclusion. The revised conclusion appears to now include new papers not discussed elsewhere in the manuscript. There are several reasons why this is problematic. First, the conclusion should not present new information, it should articulate a final conclusion based on the analysis of the totality of the data analyzed in the review. Secondly, the work that appears in the conclusion of the paper is not fully explained or evaluated. There is simply not enough information about the use of supplementation. It is important add more information, particularly given that the most recent study (Khazdouz et al., 2018) is both a human clinical trial and to my knowledge does not appear in any of the other numerous reviews on this topic. Thus, the data in the newly cited papers need to be discussed in more depth in the body of the paper.

<Response: We appreciate this reviewer’s helpful comments. In section 3.1. role of zinc on traumatic brain injury-induced neuronal death on page 10, we added an additional paragraph addressing the reviewer’s suggestion as follows. And in the Conclusion section on page 15, we re-wrote it.>  

<“3.1. Role of zinc on traumatic brain injury-induced neuronal death

On the other hand, recently evidence was presented that zinc chelation after brain injury could have adverse effects. Recent papers also suggested possible therapeutic effects of zinc deficiency in TBI, in contrast to zinc supplementation [154]. In particular, recent studies have suggested that zinc deficiency may be associated with alterations in MMP activity that may play a role in a variety of important functions associated with TBI, including disruption of the blood-brain barrier, inflammation and angiogenesis [155, 156]. Also, zinc chelation has been reported to inhibit TBI induced neurogenesis [147].

In animal models, zinc supplementation may be an effective treatment option to improve behavioral outcomes, such as cognitive impairment and depression, following TBI in rat models [157]. Khazdouz M et al. (2018) recently confirmed the efficacy of zinc supplementation in patients with severe head trauma in a double-blind controlled study. In this trial, 100 patients with severe TBI were randomly assigned to a zinc supplement or a placebo group and evaluated for 16 days. There was no difference between the two groups at baseline, but interestingly, zinc supplementation groups showed significantly increased plasma zinc concentrations. For inflammation factors, zinc supplementation group had significantly lower plasma c-reactive protein (CRP), erythrocyte sedimentation rate (ESR) and white blood cell (WBC) compared to controls on day 16. At the same time point (day 16), the assessment of neurological recovery results, zinc supplementation group had significantly lower sequential organ failure assessment (SOFA) scores than in the placebo group. Zinc supplementation group also had a significantly improved Glasgow Coma Scale (GCS) rating compared with the placebo patients group [158]. Based on these recent studies, there is ample evidence to support zinc supplementation as having a potentially beneficial role for patients with conditions associated with an acute drop in plasma zinc levels, such as head trauma.”>

<“4. Conclusion

Zinc is a trace metal ion in the central nervous system that plays three major biological roles in catalysis, structure, and regulation. Zinc contributes to antioxidant function and the proper functioning of the immune system and plays an important role in neurophysiology, ranging from cell growth to cell proliferation. It also plays a critical pathophysiological role in major neurological diseases. Zinc homeostasis is important, representing a point of intersection in related brain diseases. In particular, excessively released and accumulated zinc ions after brain insults cause neurotoxic damage to postsynaptic neurons. On the other hand, zinc deficiency causes neurodegenerative and cognitive decline disorders, such as decreased neurogenesis and impaired learning and memory.

For these reasons, the concept of preventive treatment, in which a disease related to zinc accumulation is prevented by reducing the level of zinc through a metal-chelator, is a promising method that needs further exploration. Zinc-chelating agents can bind to metal ions and form complex structures that are easily excreted by the body to remove them from the intracellular or extracellular spaces. However, these metal chelates have side effects not only on zinc concentrations but also on other important metals [200-202]. In addition, zinc deficiency could be associated with DNA fragmentation, and increased cell death and apoptosis at the site of injury after head trauma. And it impairs cell proliferation and survival through several signaling pathways (such as p53) in the brain [2, 203-205].

Conversely, in the case of degenerative disorders due to zinc deficiency such as TBI, zinc supplementation through dietary intake may be a beneficial therapeutic concept. In light of therapeutic zinc supplementation, it improved the rate of neurologic recovery and visceral protein concentrations and in reducing mortality and GCS recovery in patients with severe brain injury [206]. However, it is necessary to assess the optimal response by determining the effective dosage, timing, and therapeutic outcomes for zinc treatment in head injury patients. The administration of zinc can potentially interfere with copper absorption and can cause anemia, neutropenia, and copper deficiency associated with neurological symptoms [207-209].

Therefore, a very important aspect of understanding how zinc can be used as a therapeutic agent is the necessity of a detailed understanding of the factors that influence zinc ion homeostasis and the optimization of the relevant concentrations.”>

<Additionally, I wrote a concluding sentence for each paragraph as indicated in <Sky Blue Highlights>.